# IntelLLM: Little Hints Make a Big Difference for LLM KV Cache Compression

## Abstract

Large Language Models (LLMs) have demonstrated exceptional capabilities in integrating contextual knowledge, but their deployment is often constrained by the substantial computational resources required for long text sequences. To mitigate the inference time cost associated with attention mechanisms, LLMs utilize key-value embedding caching techniques (KV cache), which introduce significant storage pressure. In this paper, we propose IntelLLM, a novel and efficient approach to KV cache compression that strikes a balance between compression rate and performance. Drawing inspiration from sparse attention mechanism, we observe that only a small subset of tokens in lengthy texts capture the majority of attention weights. This sparsity, intrinsic to the attention mechanism, serves as the foundation for improving the KV compression ratio through a strategic eviction method. IntelLLM is composed of center of gravity eviction (CGE) strategy and remote gap localization (RGL) strategy. CGE is designed to address the potential loss of important semantic dependencies when evicting high-sparsity tokens, which prioritizes the retention of key tokens by shielding the center of gravity of attention during inference, thereby preserving critical information and optimizing the efficiency of attention computation. Additionally, RGL is proposed to leverage implicit positional features to maintain long-range dependencies, inspired by advancements in location encoding research. Our KV compression approach integrates seamlessly with existing LLMs, requiring minimal code modifications without the need for fine-tuning or model parameter changes. IntelLLM not only significantly reduces the storage requirements for KV cache but also consistently outperforms full KV models in long text processing tasks, while utilizing only 50% of the typical KV cache expenses.

## 1 Introduction

Large Language Models (LLMs) have rapidly expanded from academic research to industrial applications Wu et al. (2023)Zhao et al. (2024a) due to their exceptional capabilities in language understanding and generation, knowledge retention and retrieval, and multi-task learning Ge et al. (2024). However, as LLM applications evolve, the associated computational costs have become a critical challenge. During autoregressive inference in LLMs, the quadratic complexity of the transformer architecture and the large number of model parameters result in significant computational overhead Zhou et al. (2024). To avoid recomputation, KV caches are preserved. However, the size of the KV cache grows proportionally with the length of the inference sequence, eventually surpassing the model's parameter size as the sequence accumulates, with the size of the KV cache emerging as a key issue. This raises a significant challenge for LLM deployment: how to reduce the memory consumption of the KV cache without compromising model accuracy.

To tackle this challenge, Hooper et al. (2024b)Ye et al.Zhang et al. (b) have proposed various strategies to optimize KV caching and reduce memory consumption during inference. One prominent approach is cache compression Liu et al. (2024), which minimizes storage requirements by eliminating redundant information in the KV cache. Low-rank approximation techniques Zhang et al. (a) have been employed to compress the original high-dimensional KV representations into lower-dimensional spaces. Additionally, sparse attention mechanisms Liu et al. (2023)Fu et al. have been introduced to selectively retain only the most critical key-value pairs, discarding less relevant information and reducing both computation and cache space requirements. However, these methods

present two main limitations: firstly, there is a trade-off between memory savings and accuracy, making it difficult to strike the right balance; secondly, the added complexity of these strategies often leads to a conflict between increased computational overhead and memory optimization. Meanwhile, the windowed attention approach Beltagy et al. (2020) has been proposed as a solution to reduce KV caching costs during inference. While this approach effectively reduces memory consumption, it suffers from significant accuracy degradation when handling sequences that exceed the predefined memory window size. Comparable to the fixed window design, numerous studies have explored reverse thinking, which involve adapting long texts to a preset window size. This can be viewed as a hierarchical strategy that condenses the text, extracting only the core information to alleviate KV cache pressure Song et al. (2024)Shao et al. (2024)He et al. (2024). These methods offer valuable insights for our research, providing new perspectives on how to balance cache compression with maintaining model performance.

Inspired by the above research insights, we propose the hypothesis that in long-text reasoning tasks, LLMs may process information similarly to how humans read and comprehend lengthy articles by relying primarily on key hints. In other words, the KV cache may only need to store the most important tokens from the redundant text. Through statistical analysis, we identified a remarkably high degree of sparsity in the attention layer. This finding validates our hypothesis that LLMs heavily depend on key cues to achieve their superior long-text memory processing capabilities. It also suggests that much of the redundant information in long texts may be superfluous for comprehension and can be effectively optimized out of the KV cache.

In this paper, we introduce IntelLLM, a lightweight framework tailored for long-text inference in LLMs. IntelLLM incorporates center of gravity eviction (CGE) strategy and remote gap localization (RGL) strategy, two novel techniques that effectively balance cache compression and model performance, all without fine-tuning. In Section 3, we analyze the shortcomings of the sliding window mechanism in handling long-sequence inference tasks, focusing on the behavior of the attention layer and providing detailed insights into the attention mechanism's characteristics. Based on the problem analysis and findings in Section3, Section4 introduces the KV cache eviction and update algorithm for IntelLLM, elaborating on the center of GE strategy and the RGL strategy to mitigate performance loss. In Section5, we present the experimental validation, where IntelLLM is integrated with Llama-3-8B-instruct and Mistral-7B-inst-v0.2, and evaluated using LongBench. Without fine-tuning, IntelLLM achieves performance comparable to baseline models while significantly reducing cache memory usage. Its lightweight design also ensures easy integration into any existing LLM system.

## 2 RELATED WORK

### 2.1 LLM LONG TEXT REASONING

The significant computational and memory demands during LLM inference pose critical challenges for deployment in resource-constrained environments, limiting the capacity of LLMs to efficiently process long texts. Prior research has predominantly focused on two areas: long-text compression Zhao et al. (2024b) and model compression strategies Ramesh et al. (2023). Long-text compression techniques, such as scalable embeddings, prompt compression, and activation beacons, aim to reduce storage overhead for extended sequences. Meanwhile, model compression strategies focus on optimizing the attention mechanism, either by designing more efficient architectures or by compressing the key-value (KV) cache to alleviate memory pressure. In this paper, we address the bottlenecks in LLM inference, with a particular focus on optimizing the KV cache.

### 2.2 QUANTIZATION

For parameter and computation intensive tasks in LLMs, quantization techniques have proven to be highly effective in reducing memory consumption and accelerating inference. Current research primarily focuses on enhancing computation-heavy operations through low-bit integer quantization of model weights. For instance, Wu et al. (a) employs INT4 quantization for both weights and caches, significantly reducing GPU memory requirements. Additionally, advanced hyper-quantization methods such as KVQuant Hooper et al. (2024a) and WKVQuant Yue et al. (2024) have been introduced to further optimize memory usage. However, while these quantization techniques succeed in com-

pressing models from a parametric standpoint, they do not fully address the KV cache pressure inherent in long dialogue history inference tasks. This gap highlights the need for more comprehensive strategies to tackle the specific challenges of long-context processing in LLMs.

### 2.3 KV COMPRESSION / SPARSE ATTENTION

KV cache optimization has become a pivotal research area due to the substantial computational burden imposed by the quadratic complexity of the attention mechanism. Techniques such as clustering-based hashing Kitaev et al. (2020) and k-nearest neighbors (kNN) Nawrot et al. (2024) methods effectively reduce computational complexity to super-linear levels. Moreover, the introduction of sparse Transformers improves computational efficiency through sparse attention mechanisms. Approaches like MQA Shazeer (2019) and GQA Ainslie et al. (2023) for instance, optimize the attention module by redesigning attention heads, while more recent methods focusing on KV cache reuse Gim et al. (2024) and cross-layer cache sharing Brandon et al. (2024) further improve memory efficiency. However, these approaches often incur additional training overheads, posing challenges for deployment in resource-constrained environments. This underscores the demand for more efficient solutions that can minimize resource consumption without sacrificing performance.

### 2.4 OFFLOADING

To address the high memory demands of KV caches on GPUs, researchers have begun exploring alternative memory strategies involving multi-cluster systems or CPUs. For instance, DistKV-LLM enhances cloud service inference by distributing KV caches across multiple servers. Similarly, Pan et al. (2024) Wu et al. (b) propose a method that offloads KV caches to CPUs, where only a small portion of the cache is reloaded to the GPU, significantly lowering the memory requirements for high-performance computing (HPC) devices. Although these approaches optimize inference by leveraging external storage, they also introduce additional memory resource requirements. In contrast, the research presented in this paper focuses on alleviating the computational burden of inference by efficiently utilizing limited memory resources, without relying on external memory expansion.

While existing compression and offloading techniques partially alleviate KV cache pressure in LLM inference, the trade-off between performance and resource consumption persists. In this paper, we introduce an optimization strategy that eliminates reliance on external memory, enabling effective KV cache compression without sacrificing model performance. This novel approach provides a robust solution for long-text reasoning in LLMs, particularly in resource-constrained environments, thereby enhancing the practical applicability of LLMs in real-world settings.

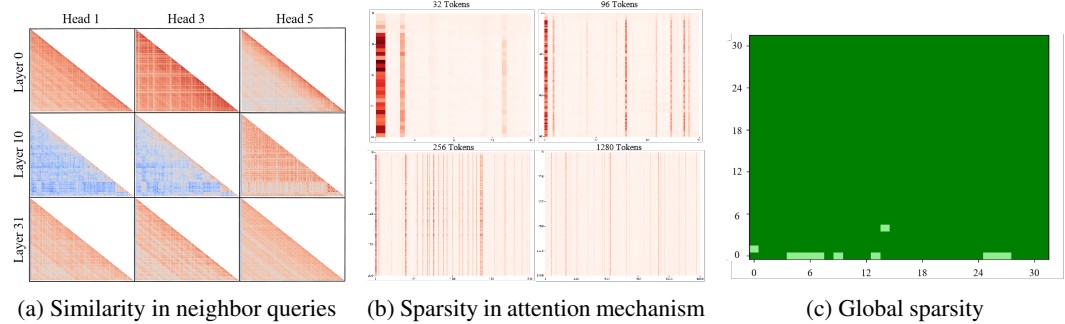

(a) Similarity in neighbor queries  (b) Sparsity in attention mechanism  (c) Global sparsity

Figure 1: (a) Visualization of Cosine Similarity for Queries Based on Llama3-7B. The nearest-neighbor queries across different attention layers and heads exhibit a high degree of similarity. (b) Visualization of attention scores across texts of varying scales. (c) Sparsity of the full attention layer. The dark green regions indicate values that exceed the threshold of 85%.

# 3 FINITE WINDOW HIGH COMPRESSIBILITY ASSUMPTION

## 3.1 PROBLEM ANALYSIS

LLM is constructed based on casual transformer, which is divided into embedding layer, feedforward layer and self-attention layer. We focus on the self-attention layer in the decoding phase.

**Extra-domain inference.** During the pre-training phase, the model establishes a maximum text length, making Softmax highly sensitive within this range. However, when covariates with sequence lengths are introduced into the self-attentive mechanism with significantly exceeding the pre-training length, they compromise the robustness of the attention distribution. This results in the distributional moderation imbalance problem in Softmax, rendering LLM inference unpredictable. As positional and semantic information cannot be easily decoupled, the pre-training window length becomes a constant challenge for extra-domain inference, especially without optimizing or fine-tuning the model architecture. We refer to this phenomenon as the problem of **out-of-domain distributional disequilibrium (ODD)**.

**Intra-domain Inference.** When the size of the sliding window is confined to the pre-training window, window attention mitigates the imbalance caused by relative positional differences in ODD. However, it still fails to reason effectively about long texts. From a semantic perspective, we contend that attention during window sliding relies entirely on short token sequences relative to the current time step. This premature discarding of long text information, coupled with the emphasis on short-term dependencies, directly contributes to the collapse of the LLM.

Consequently, we can draw two important insights:

**Conclusion 1.** Focusing solely on the attention layer, the stability of the LLM's ability to retain comprehension of long sequences is contingent upon the high sensitivity of Softmax in assigning weight streams.

**Conclusion 2.** The failure of sliding-window long-text inference arises from the limited robustness of the window's attention, indicating that the distribution of attention is heavily influenced by future markers. The query is continuously updated as the text expands, leading the model's attention to prioritize local information features at the expense of capturing contextual dependencies. This phenomenon is exacerbated by unpredictable noise in long texts, the prominence of locally relevant information, and conflicts between global features.

## 3.2 LOCAL TOKEN MODELING

Building on the relationship between in-domain and out-of-domain inference as discussed in Section3.1, we divide the long context sequence into two components based on the current time step: local tokens $L_{local}$ and remote tokens $L_{remote}$. Consequently, the long context can be denoted as $L = L_{local} + L_{remote}$. We then explore distinct compression strategies tailored to the unique features of these two components.

In the case of $L_{local}$, due to the autoregressive nature of LLM inference, the query at the current time step remains inaccessible to subsequent inference, while the query input at the next time step influences the reallocation of attention. This presents challenges in designing stable KV eviction strategies. To address this issue, we focus on uncovering solutions from the query cluster data itself.

**Research 1.** We explore the consistency characteristics of query clusters by analyzing tokens within the sequence $L_{local}$. For each head query in the same layer, we compute the cosine similarity and observe the intriguing vector distribution properties shown in Figure1a. Notably, the closer the data points are to the diagonal split line, the redder their color becomes, indicating a high similarity between adjacent queries. This observation suggests that near-neighbor queries exhibit strong contextual similarity at the semantic level. Furthermore, this phenomenon aligns with the model's reliance on prior input information to generate appropriate responses.

**Theorem 1.** In longer inference streams, retaining a limited number of clusters of near-neighbor tokens helps stabilize normal inference for local short texts.

## 3.3 REMOTE TOKEN MODELING

For distant contexts, inspired by the success of sparse attention, we hypothesize that certain key tokens remain relevant for future reasoning in long-text inference. In other words, the LLM's comprehension and memory capabilities depend primarily on these key tokens, while irrelevant token clusters that appear in long texts serve only to support localized short-text reasoning and can be treated as inactive information suitable for eviction. If this assumption holds, it would significantly alleviate the memory pressure on the KV cache.

**Research 2.** We explore the relationship between key tokens and LLM capabilities in processing long texts through an analysis of attention scores. Focusing on the compressibility of distant contexts, we emphasize the importance of long-range dependencies within the attention mechanism. As shown in the Figure1b, we visualized attention scores from NarrativeQA's scene quizzes. A notable observation is the consistent alignment of red bands within the long-sequence attention scores, indicating that queries maintain coherence around specific tokens in lengthy historical texts. Additionally, the important tokens receiving high attention scores are highly sparse.

To further validate the pervasive sparsity of these high-scoring tokens, we set the attention threshold at time step $t$ to $1/t$, considering only attention scores exceeding this threshold as important key locations of focus for the query. As depicted in the Figure1c, both individual attention heads and entire attention layers demonstrate significant sparsity, with over 90% of the attention being sparse. This observation reinforces our hypothesis that key tokens in distant long-text contexts are highly sparse and exert a lasting impact on the model's reasoning over extended sequences.

**Theorem 2.** For remote tokens, queries exhibit high attention to only a few key historical messages, with highly scored keys displaying significant sparsity across both the attention layers and heads. By retaining only the important key-value pairs, the long-text processing capabilities of the LLM can be preserved without loss, enabling a high degree of compression for distant tokens.

## 4 INTELLLM

Based on the theorems from Section3, this section introduces IntelLLM, a novel approach that eliminates the need for fine-tuning while significantly reducing the memory demands of the KV cache and improving the LLM's generalization capability for longer sequences. In Section4.1, we present the CGE strategies, designed based on two distinct characteristics of language task data. Then Section4.2 details the design of the RGL strategy, drawing inspiration from positional encoding research. Furthermore, Section4.3 outlines the windowing mechanism implemented in IntelLLM.

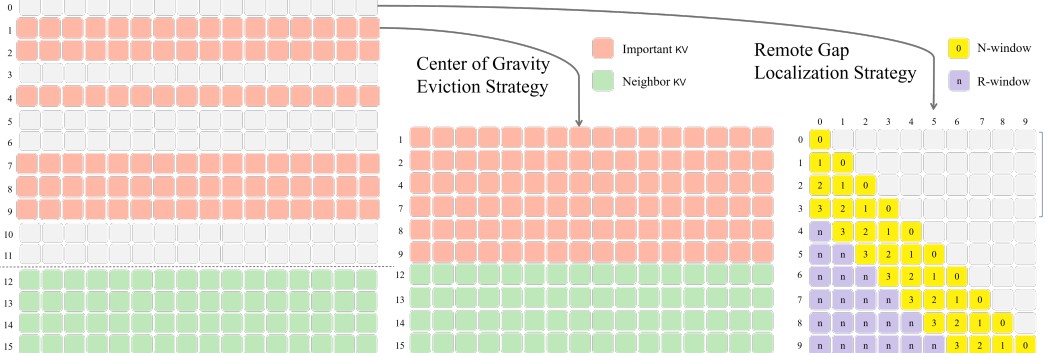

Figure 2: IntelLLM composed of CGE and RGL.

## 4.1 CENTER OF GRAVITY EVICTION

In attention mechanisms, softmax functions are commonly employed to compute weights that guide the selection and focus of information. As discussed in Section3.1, softmax is highly sensitive to attentional matching within the pre-training window. However, when we compress the KV cache by

evicting redundant KVs, the tokens that accumulate within this window become a center of weight concentration, due to their importance in long-text processing.

$$SoftMax(x)_i = \frac{e^{x_i}}{\sum_{j=1}^{n} e^{x_j}} \tag{1}$$

From a theoretical perspective, when a significant accumulation of weights occurs, we can assume that the center of gravity of attention is represented by $x_k$, $x_k$ represents the clustering of important KVs, with its attention score significantly exceeding that of other tokens. Based on the properties of softmax, we can simplify the output by considering the limit as:

$$SoftMax(x)_i \approx \begin{cases} 1, & \text{if } i = k \\ 0, & \text{if } i \neq k \end{cases} \tag{2}$$

This implies that the softmax output will be heavily concentrated in the region corresponding to the maximum weight, i.e., the cumulatively important keys, effectively rendering the attention scores of other tokens close to zero.

When softmax becomes imbalanced, with large weights dominating the output, the attention mechanism may lose focus on other critical information. This imbalance can reduce the model's representation capacity, as it fails to fully leverage all available input data.

To restore the balance of the softmax function and mitigate the inference instability caused by distant context compression, we adopt a strategy of evicting the attention center of gravity, which prevents large weight dominance and further stabilizes the distribution of attention scores. Based on Equation 1, the following formula can be derived:

$$SoftMax(x)_i = \frac{e^{x_i - x_k}}{\sum_{j=1}^{n} e^{x_j - x_k}} = \frac{e^{x_i - x_k}}{1 + \sum_{j \neq k} e^{x_j - x_k}} \tag{3}$$

when $x_i = x_k$, $e^{x_i - x_k} = e^0 = 1$. As $x_j \ll x_k$, $e^{x_j - x_k} \to 0$. The CGE strategy effectively mitigates the imbalance caused by dominant large weights by redirecting attention away from the center of gravity.

At the same time, based on the principle of weight accumulation, we focus the attention's center of gravity on two key regions. As shown in the Figure1b, a small number of tokens at the beginning consistently accumulate a significant portion of the attention scores, forming **the head gravity**. Additionally, as discussed in Section3.2, the high similarity of near-neighbor queries leads to geometric accumulation of attention scores. Thus, the second region of interest corresponds to **the tail gravity**, which concentrates on these similar near-neighbor query clusters.

## 4.2 REMOTE GAP LOCALIZATION

During the attention computation process, both the query and the key originate from the input sequence. By calculating the attention scores between them, the model identifies similarities, allowing it to focus on the most relevant information at each time step. The attention scores are therefore positively correlated with the degree of focus. Typically, without explicit positional encoding (e.g., absolute or relative position), the attention mechanism is not inherently sensitive to positional distance and relies solely on query-key similarity to highlight pertinent content.

We aligned the positional encoding with the compression window size and leveraged relative positional differences to represent the relationship between the compressed historical long text and the current query. However, our experimental results were less than satisfactory. So we hypothesize that in the next time step of inference, simply reassigning positions is insufficient to capture the attentional relationship between the current query and the compression window, even if the relative positional distance remains within the pre-training length. As reasoning progresses, the information in the compression window spans much further than the current time step. Thus, the temporal structure of the sequence at the semantic level remains intact, even when the KV cache is compressed to fit within the pre-training window, and cannot be fully represented by a simple approximation of

---

**Algorithm 1:** Compression algorithms

---

**Input:** Last Query $Q \in R^{n \times d}$, Key Cache $\hat{K} \in R^{m \times d}$, Value Cache $\hat{V} \in R^{m \times d}$, Head Gravity
     Len $l_{head}$, Tail Gravity Len $l_{tail}$

1   $A^0 \leftarrow QK^T / \sqrt{d}$                                      ▷ Attention score of $qk^T$

2   *# Global dependency algorithms*

3   $A^1 \leftarrow softmax(A^0_{[:l_{head}]})$             ▷ Masking head gravity of during normalization

4   $A^2 \leftarrow sum_v(A^1_{[:l_{tail}]})$                ▷ Masking tail gravity, sum along vertical

5   *# Local fine-grained modeling algorithms*

6   $A^1 \leftarrow softmax(A^0_{[:l_{tail}]})$              ▷ Masking tail gravity of during normalization

7   $A^2 \leftarrow sum_v(A^1_{[:l_{head}]})$               ▷ Masking head gravity, sum along vertical

8   $Indices \leftarrow argsort(argtopk(A^2))$        ▷ Ranking indices of the top k positions

9   $K_{comp}, V_{comp} \leftarrow gather(Indices)$      ▷ Key clusters within the compression window

10   $\hat{K}, \leftarrow concat(K_{comp}, \hat{K}_{[-l_{tail}:]})$

11   $\hat{V} \leftarrow concat(V_{comp}, \hat{V}_{[-l_{tail}:]})$

12   $return \hat{K}, \hat{V}$           ▷ Integration of compression window and neighbor window

---

positional differences. This inherent positional relationship continues to influence the distribution of attention scores.

To address this, we propose the Remote Gap Localization (RGL) strategy, as shown in Figure2. The RGL strategy primarily addresses the issue of time span vanishing caused by high compression by assigning cache position values significantly larger than the length of the near-neighbor window to distant KVs within the compression window. Meanwhile, it preserves the relative position information of the near-neighbor window, ensuring stability in capturing long-range dependencies..

### 4.3 DESIGN

Leveraging the CGE and RGL strategies, IntelLLM incorporates two primary elements: the compression window and the nearest-neighbor window, as illustrated in Figure2.

#### 4.3.1 COMPRESSION WINDOW

In conjunction with the CGE strategy, we stabilize KV eviction by masking the influence of the center of gravity during the normalization process. However, through our experiments, we found that the combination of the masking order of the attention center of gravity and the normalization operation has different impacts on model performance. Based on these experimental observations, we categorize them into two types of characterization algorithms, providing explanations for each in relation to the semantic features of specific textual tasks.

**Local Fine-grained Modeling Tasks.** Compared to tasks requiring long-term global dependencies, this task focuses more on modeling fine-grained local information in the current context. In natural language processing, short-term dependencies are typically closely tied to the immediate context. For example, in conversational tasks, information from neighboring time steps has a greater impact on the model's predictions than distant contexts. Consequently, the model emphasizes the semantic features of nearby tokens during inference. As shown in Algorithm1, we implement a local focus of attention by first masking the head gravity according to the short-term dependency strategy, followed by removing the effect of the tail gravity after data normalization.

**Global Dependency Tasks.** Since this kind of task focuses more on requiring the model to rely on long-distance contextual information in the inference process, the compression strategy in this phase should focus more on the distant above. Therefore, as illustrated in Algorithm1, we first shield the influence of the tail gravity before the normalization operation, followed by masking the weaker influence of the head gravity during the selection of important keys. This approach effectively enhances the model's focus on distant information while emphasizing the consistency and completeness of global semantics.

### 4.3.2 Nearest Neighbor Window

Based on the findings in Section3.2, the nearest-neighbor window stabilizes the inference process for subsequent queries by retaining the tokens in close proximity. This approach ensures that the model maintains focus on relevant nearby information, supporting more accurate and stable predictions.

## 5 Evaluation

In this section, we demonstrate through validation experiments that IntelLLM achieves KV cache compression using a simple window combination. This approach effectively compensates for long text inference accuracy while delivering exceptional performance across a wide range of domain tasks.

### 5.1 Experimental Settting

To comprehensively evaluate IntelLLM's performance in long text reasoning tasks, we employ LongBench for long document benchmarking. To validate the rationale behind the IntelLLM architecture design, we employ the original model with full KV caching as a strong baseline. Since IntelLLM aims to balance performance and KV compression without requiring fine-tuning, we also include two windowed approaches LM-Infinite and StreamingLLM as additional baselines. All evaluation experiments are conducted on a single NVIDIA A100 80GB GPU server.

Table 1: Comparison of model performance based on Llama3. The baseline results, marked with *, are reproduced from Xiao et al.. All baseline models utilized the pre-training window size as their context length, while IntelLLM is configured with a total window length of 4K ($L_{comp} = L_{near} = 2K$). Refer to the AppendixA for corresponding data IDs and names.

| Llama-3 | Window | Single-Doc QA | | | Multi-Doc QA | | | Code | |
|---|---|---|---|---|---|---|---|---|---|
| | | 1-1 | 1-2 | 1-3 | 2-1 | 2-2 | 2-3 | 3-1 | 3-2 |
| Full* | 8K | 19.85 | 42.36 | 41.03 | 47.38 | 39.2 | 22.96 | 60.83 | 49.14 |
| Infinite* | 8K | 19.39 | 42.80 | 40.44 | 43.77 | 37.89 | 18.33 | 60.12 | 48.62 |
| Streaming* | 8K | 20.05 | 42.46 | 39.54 | 43.69 | 37.89 | 19.68 | 60.35 | 48.95 |
| IntelLLM* | 4K | 22.16 | 40.88 | 48.06 | 44.3 | 35.2 | 22.74 | 58.39 | 53.76 |

| Llama-3 | Window | Few-shot Learning | | | Synthetic | Summarization | | |
|---|---|---|---|---|---|---|---|---|
| | | 4-1 | 4-2 | 4-3 | 5-1 | 6-1 | 6-2 | 6-3 |
| Full* | 8K | 74 | 90.5 | 42.3 | 8.5 | 29.94 | 21.45 | 27.51 |
| Infinite* | 8K | 74 | 90.08 | 41.72 | 4.5 | 29.25 | 21.41 | 27.62 |
| Streaming* | 8K | 73.5 | 90.08 | 41.55 | 5 | 29.17 | 21.33 | 27.56 |
| IntelLLM | 4K | 73.5 | 90.7 | 42.21 | 7 | 30.13 | 21.62 | 27.96 |

### 5.2 Main Results

We select the Llama-3-8B-Instruct model for a comprehensive evaluation on the LongBench dataset, with the corresponding test results presented in Tables1. Our data analysis leads to the following conclusions: (1) Compared to the baseline streaming input model, IntelLLM's token eviction strategy demonstrates superior performance, indicating that IntelLLM enhances the LLM's understanding of long sequential contextual information through key hints, thus maintaining the stability of streaming inference; (2) While the two windowing mechanisms efficiently conserve KV cache space, they struggle to effectively capture long-range text dependencies. IntelLLM mitigates the performance degradation caused by compression through its adaptive tuning mechanism for the compression windows, offering a partial solution to this challenge; (3) With 50% KV cache compression, IntelLLM achieves performance close to or even exceeding that of the original strong baseline, validating the effectiveness of our strategy while significantly reducing memory costs and enabling long sequence reasoning in LLMs with low-resource utilization.

The results of our evaluation of Mistral-7B-inst-v0.2 are shown in Table3, which also indicates that IntelLLM outperforms other approaches.

Table 2: Comparison of model performance based on Mistral. The baseline results, marked with *, are reproduced from Xiao et al..IntelLLM is configured with a total window length of 4K ($L_{comp} = L_{near} = 2K$).

| Mistral | Window | 1-1 | 1-2 | 2-1 | 4-1 | 4-2 | 4-3 | 5-1 | 6-1 |
|---------|--------|-------|-------|-------|-------|------|-------|-------|------|
| Full* | 32K | 22.06 | 47.65 | 21.96 | 26.62 | 71 | 85.97 | 42.29 | 3.95 |
| Infinite* | 6K | 18.44 | 39.05 | 22.27 | 26.65 | 70 | 85.22 | 41.6 | 2.08 |
| Streaming* | 6K | 17.92 | 39.09 | 21.83 | 26.64 | 70 | 85.57 | 41.31 | 2.5 |
| IntelLLM | 4K | 19.19 | 47.11 | 21.59 | 26.63 | 70.5 | 86.81 | 41.67 | 2.87 |

**Latency.** Based on Llama3, we measure the inference latency for 8K text sequences. In comparison to the full-cache model inference latency of 900.84 ms, the KV update algorithm of IntelLLM adds only an additional 2.37 ms. Considering the achieved 50% cache savings, this 2.63% increase in latency is entirely acceptable.

**Ablation Study.** To validate the impact of IntelLLM's two policies on long text processing tasks, we select various task types and present the results of the ablation study for the Llama-3-8B-Instruct model in Table3. Given the inherent negative correlation between KV cache compression and performance, we use CGE and RGL as ablation terms to illustrate their effects on performance under limited KV caching conditions.

Table 3: Ablation with CGE and RGL.

| IntelLLM | Head Gravity | Tail Gravity | RGL Gap | Result |
|----------|--------------|--------------|---------|--------|
|          | 0            | 2K           | 4k      | 21.4   |
| **1-1**  | 4            | 2K           | 4K      | 21.04  |
|          | 6            | 2K           | 4k      | 22.16  |
| **IntelLLM** | **Head Gravity** | **Tail Gravity** | **RGL Gap** | **Result** |
|          | 4            | 2K           | 4k      | 30.94  |
| **2-2**  | 4            | 2K           | 6K      | 31.83  |
|          | 4            | 2K           | 8k      | 35.2   |

**CGE Ablation.** CGE enhances the model's attentional focus by masking different attentional anchors, facilitating the management of both long and short dependencies across various task types. In short-term dependency tasks, we fix the positional interval between the tail center of gravity and the compression window, and assess the effectiveness of the CGE strategy by scaling the head center of gravity region. As shown in Table3, the weight distribution within the head center of gravity clusters has a significant impact on the model's performance in processing dialog tasks.

**RGL Ablation.** In Table3, we evaluate the correlation between RGL's positional distance interval settings and the model's inference performance. Given the constraints of the model's pre-training window, we limit the relative positional differences to fall within this range. The results show that using positional intervals to represent the semantic distance or time span between the nearest-neighbor window and the salient window proves to be an effective approach.

## 6 CONCLUSION

In this paper, we tackle the memory challenges associated with KV caching in LLM deployments by introducing IntelLLM. Our extensive evaluation experiments demonstrate that IntelLLM significantly improves performance through the combined use of CGE and RGL strategies. This approach allows LLMs to achieve an optimal balance between KV cache compression and inference performance without requiring fine-tuning, all while avoiding substantial overhead in computational resources. As a result, IntelLLM effectively enhances the ability of LLMs to reason about long-text tasks.

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

# A APPENDIX

Table 4: Dataset ID Mapping Table

| Dataset | ID | Source | Metric |
|---|---|---|---|
| *Single-Doc QA* | | | |
| NarrativeQA | 1-1 | Literature,Film | F1 |
| Qasper | 1-2 | Science | F1 |
| MultiFieldQA | 1-3 | Multi-field | F1 |
| *Multi-Document QA* | | | |
| HotpotQA | 2-1 | Wikipedia | F1 |
| 2WikiMultihopQA | 2-2 | Wikipedia | F1 |
| MuSiQue | 2-3 | Wikipedia | F1 |
| *Code Completion* | | | |
| LCC | 3-1 | Github | Edit Sim |
| RepoBench-P | 3-2 | Github repository | Edit Sim |
| *Few-shot Learning* | | | |
| TREC | 4-1 | Web question | Accuracy |
| TriviaQA | 4-2 | Wikipedia, Web | F1 |
| SAMSum | 4-3 | Dialogue | Rouge-L |
| *Synthetic Task* | | | |
| PassageCount | 5-1 | Wikipedia | Accuracy |
| *Summarization* | | | |
| GovReport | 6-1 | Government report | Rouge-L |
| QMSum | 6-2 | Meeting | Rouge-L |
| MultiNews | 6-3 | News | Rouge-L |

