# OpenReview forum: "IntelLLM: Little Hints Make a Big Difference for LLM KV Cache Compression"
_ICLR.cc/2025/Conference — ICLR 2025 Conference Withdrawn Submission_

### Official Review · Reviewer_4hs9 · 2024-10-16

**Soundness:** 2
**Presentation:** 2
**Contribution:** 3
**Rating:** 3
**Confidence:** 4

**Summary:**

This paper presents IntelLLM, a novel KV cache eviction algorithm designed to alleviate the storage and computational burden of Transformer-based large language model inference. IntelLLM leverages the sparsity of attention mechanisms and strategically evicts certain tokens to improve inference efficiency. The authors propose two key strategies: Center of Gravity Eviction (CGE) and Remote Gap Localization (LGL). CGE addresses the semantic loss caused by dominant attention scores in softmax. LGL reorganizes token positions by creating a large gap between global and local tokens, further enhancing processing efficiency. IntelLLM is evaluated on LongBench with two models, Llama-3-8B and Mixtral-7B-instruct-v0.2, and outperformed all the baselines, including Full KV cache, StreamingLLM, and LM-Infinite. The algorithm achieved 2x KV cache compression ratio for Llama and an 8x compression ratio for Mixtral, maintaining strong inference efficiency despite the reduced cache size. Additionally, the paper includes ablation studies focusing on Head Gravity and the RGL Gap, demonstrating the soundness and effectiveness of IntelLLM's design.

**Strengths:**

1. This paper provides a clear explanation of why the attention mechanism tends to allocate excessive weights to a few tokens with high attention scores while neglecting others. The analysis of the softmax function is insightful and well-reasoned.

2. The proposed method, IntelLLM, demonstrates strong performance in benchmarks and shows compatibility across multiple models, including widely-used ones like Llama and Mixtral.

3. The method is simple and easy to implement. IntelLLM can be integrated into various scenarios and inference frameworks.

**Weaknesses:**

1. The presentation is to some extent unclear and confusing. Some notations and terminologies are used without a clear definition. The description of Algorithm 1 is confusing and hard to understand. Tables and figures and be organized in a better format. Details of presentation issues are described in question Q1.

2. Insufficient baseline selection. The proposed algorithm of this paper is an eviction algorithm that compresses KV cache, thus there should be comparison between other popular eviction methods, including H2O [1], SnapKV [2], SirLLM [3] and IntervalLLM (a baseline created in SirLLM). It would be better to compare IntelLLM with other end-to-end methods, including InfLLM [4] and MInference [5], as they utilize attention sparsity. (This is only a suggestion and the authors are not required to test all the methods above during the review process, but futher discussion is expected.)

3. Insufficient benchmark datasets. Although LongBench is a classic benchmark for long context inference, the average length is relatively short and the hardness is limited. It is expected to benchmark IntelLLM on harder benchmark, e.g. L-Eval [6], and longer benchmark, e.g. RULER [7] or InfiniteBench [8]. It would also be better to test IntelLLM on accurate context retrieval tasks, e.g. Ret.PassKey in InfiniteBench and RULER. (This is also a suggestion, and the authors are not required to test all of the benchmarks mentioned above. However, some simple supplementary benchmarks are welcomed.)

4. Lack of ablation studies. Ablations on $L_{comp}$ and $L_{near}$ are expected, as they can be used to prove which part of the retained KV cache is more important and to what extent can the hyperparameters affect the overall performance. Also, there should be comparison between RGL and other methods of assigning position information, such as assigning continuous position ids and not assigning any position information for distant tokens.

5. Unclear presentation of research intention. KV cache compression is a technique developed to enhance model generation speed or reduce memory consumption. The paper should clearly state out which purposed is focused mainly, and conduct the corresponding experiments. For example, the model generation speed should be tested and it is expected to be faster than full KV cache inference. The peak memory is also expected to be lower if the system implementation is delicate enough. The speed reported in line 445-446 cannot state the research intention as the pre-fill speed is slightly lower. Reporting a faster generation speed (the speed only tested on the decoding stage and excluding the pre-fill stage) might be helpful.

6. No limitation and future work discussion. This paper should include some vital limitations and potential future work of the proposed methods.


[1] H2O: Heavy-Hitter Oracle for Efficient Generative Inference of Large Language Models

[2] SnapKV: LLM knows what you are looking for before generation

[3] SirLLM: Streaming Infinite Retentive LLM

[4] InfLLM: Unveiling the intrinsic capacity of LLMs for understanding extremely long sequences with training-free memory

[5] Minference 1.0: Accelerating pre-filling for long-context llms via dynamic sparse attention

[6] L-eval: Instituting standardized evaluation for long context language models

[7] RULER: What’s the Real Context Size of Your Long-Context Language Models?

[8] ∞ Bench: Extending Long Context Evaluation Beyond 100K Tokens

**Questions:**

1. Suggestion on improving the overall presentation. First, there are some notation used without definition, e.g. $n, m, l_{head}, l_{tail}$ in Algorithm 1, and the operation $[:]$ (which dimension is applied to?) used in Algorithm 1. Second, terms such as "Center of Gravity" should be defined formally in the KV cache eviction topic. Third, there remains some minor grammar and expression mistakes in this paper, such as the misuse of citation format. Forth, the presentation of the tables and figures should be improved. The average score should be reported in Table 1 & 2. The meaning of the x-axis and y-axis of Figure 1 & 2 should be clarified.

2. The description of Algorithm 1 is confusing. Could you provide a very detailed explanation of the calculation process in natural language?

3. There isn't any model named "Llama3-7B". Do you mean Llama-3-8B?

4. What are the detailed settings of the baselines, e.g. how much KV cache is retained? Could you provide more information on the hyperparameters used on baselines?

5. The experimental results mentioned in line 316-319 should be included in this paper if such conclusion is drawn. If there is no space, it should be placed in the Appendix. The experiments in line 359-361 should also be included.

Overall, I really like the idea proposed by this paper, which I personally find very inspiring. I will raise my rating on soundness and overall rating if good discussion is made with necessary experimental results during the rebuttal phase.

---

### Official Review · Reviewer_4QgB · 2024-11-02

**Soundness:** 1
**Presentation:** 1
**Contribution:** 2
**Rating:** 3
**Confidence:** 4

**Summary:**

The paper presents IntelLLM, a method for compressing the key-value (KV) cache in Large Language Models (LLMs) to address memory constraints in long-sequence processing. Drawing on the sparsity of attention mechanisms, IntelLLM focuses on retaining only essential tokens, significantly reducing the KV cache size without compromising model performance. The proposed approach combines two strategies: center of gravity eviction (CGE), which prioritizes important tokens to preserve key semantic information, and remote gap localization (RGL), which maintains long-range dependencies using positional features. IntelLLM can integrate smoothly with existing LLMs, requiring minimal modifications and no fine-tuning. Experimental evaluations show IntelLLM achieves performance close to StreamLLM while halving KV cache requirements.

**Strengths:**

The problem investigated in the paper, KV cache compression, is crucial for long-context generation.

**Weaknesses:**

a.	The paper lacks substantial references from the past year, missing important studies like SnapKV[1], FastGen[2], H2O[3], PyramidKV[4] and etc.
b.	The observation that only a limited subset of tokens is critical for long-context generation has been extensively discussed in these and other recent works, which should be cited to provide a more comprehensive background.
c.	The experimental baseline used in the study is relatively weak; including stronger baselines from the above-mentioned works would enhance the robustness of the comparative analysis and strengthen the validity of the results.
d.	In Section 3, the two presented "theorems" are more accurately findings, as no formal proofs are provided to substantiate these claims.

[1] SnapKV: LLM Knows What You are Looking for Before Generation
[2] Model Tells You What to Discard: Adaptive KV Cache Compression for LLMs
[3] H2O: Heavy-Hitter Oracle for Efficient Generative Inference of Large Language Models
[4] PyramidKV: Dynamic KV Cache Compression based on Pyramidal Information Funneling

**Questions:**

There is no reference to table 2, why the experiments are different on Mistral and Llama?

---

### Official Review · Reviewer_BUQq · 2024-11-02

**Soundness:** 1
**Presentation:** 1
**Contribution:** 1
**Rating:** 3
**Confidence:** 3

**Summary:**

This paper aims to further the Pareto frontier of KV-cache compression rate and performance. By employing strategic eviction strategies, the method leverages the observation that only a small subset of tokens in long texts capture the majority of attention weights.

**Strengths:**

**Significance:** The paper claims 50% KV cache compression without a significant drop in performance, outperforming other KV cache compression methods in a majority of datasets in LongBench and outperforming full KV cache in some datasets. The method does not require fine-tuning, making it easy to apply to existing LLMs.

**Weaknesses:**

1. There does not seem to be a rigorous proof or set of empirical observations to substantiate the theorems proposed in sections 3.2 and 3.3.
2. There is no discussion on the method used to choose "k" – the number of top keys to be treated as "centers of gravity".
3. Implementation details are not provided, making it hard to reproduce the results.
4. There does not seem to be an ablation study to evaluate how the method performs when only either CGE or RGL is used.
5. There is no discussion on specific deployment environments where a 50% memory saving will enable new use cases, or how this method can be combined with other methods to further increase memory savings.
6. The experiments only provide two KV cache compression methods as baselines, leaving out other KV cache compression methods that do not require fine-tuning, such as static prefix caching, paged attention, or radix attention. Additionally, the experiments do not compare with other approaches that do not involve KV cache compression.
7. Best performing methods are not clearly marked in the experiment result tables.
8. The explanations and visualizations of CGE and RGL are unclear.

**Questions:**

1. What is the rationale behind choosing StreamingLLM and InfiniteLLM as KV cache compression baselines?

---

### Official Review · Reviewer_TdSu · 2024-11-03

**Soundness:** 2
**Presentation:** 3
**Contribution:** 2
**Rating:** 3
**Confidence:** 4

**Summary:**

The paper introduces IntelLLM, a framework that aims to optimize the key-value (KV) cache compression for large language models (LLMs) without compromising performance. It addresses the challenge of high memory consumption during long-sequence inference by using two innovative techniques: Center of Gravity Eviction (CGE) and Remote Gap Localization (RGL). CGE prioritizes important tokens in attention mechanisms to ensure efficient memory use, while RGL preserves essential long-range dependencies using positional features. These strategies enable significant memory savings, reducing KV cache usage by 50%, with only a minimal impact on inference latency. The authors demonstrate IntelLLM's effectiveness through comprehensive experiments, achieving performance comparable to or better than full KV models.

**Strengths:**

The combination of CGE and RGL provides a novel solution to the KV cache memory challenge, enhancing memory efficiency without fine-tuning or substantial performance loss. The paper presents a strong theoretical basis for its methods, including insights into the sparsity of attention weights and the impact of key tokens, strengthening the validity of the proposed strategies. IntelLLM is easy to integrate into existing LLM frameworks, as it requires minimal modifications, making it highly practical for real-world deployment, especially in resource-constrained environments. The extensive benchmarking on LongBench with models like Llama-3-8B-Instruct and Mistral-7B demonstrates IntelLLM's efficiency and adaptability across diverse tasks, validating the approach. Achieving 50% KV cache reduction with a negligible increase in latency is a noteworthy achievement, making IntelLLM suitable for long-text inference tasks in various settings.

**Weaknesses:**

1. Sparse attention is already well explored in several previous works as [1] [2]. This will weaken the novelty of this work. H2O [3] has already well-explored the feedback of using sliding window.
2. Lack of baselines (i.e., H2O [3], SnapKV [4], PyramidKV [5])
3. Evaluation of Needle in a Haystack is required to help illustrate your motivation of maintaining long-range dependencies
4. GCE is pretty close to previous methods like H2O [3] and SnapKV [4]. I can only see limited novelty over this method.

[1] https://arxiv.org/abs/2402.17762
[2] https://arxiv.org/pdf/2309.17453
[3] https://arxiv.org/abs/2306.14048
[4] https://arxiv.org/abs/2404.14469
[5] https://arxiv.org/abs/2406.02069

**Questions:**

As weakness

---

### Official Review · Reviewer_hxLH · 2024-11-04

**Soundness:** 2
**Presentation:** 3
**Contribution:** 2
**Rating:** 3
**Confidence:** 4

**Summary:**

The paper develops a KV-cache compression technique called IntelLLM to optimize inference in LLM on long text tasks. IntelLLM consists of two cache eviction strategies: center of gravity eviction (CGE) and remote gap localization (RGL). CGE mitigates the domain semantic imbalance by redirecting attention away from the center of gravity attention (cluster of important KVs). RGL solves the issue of time span vanishing caused by cache compression by assigning cache position values to distant KVs. Empirical evaluation shows IntelLLM saves KV cache memory by 50% with similar performance on long text tasks compared to baseline.

**Strengths:**

- The paper addresses an important research problem, KV cache optimization for  LLM inference, and proposes two interesting techniques: center of gravity eviction (CGE) and remote gap localization (RGL).

- Empirical results are competitive with prior work and baseline models.

**Weaknesses:**

- Using sparsity in attention to compression KV cache is not new. Two ICLR 2024 papers: StreamingLLM (https://openreview.net/forum?id=NG7sS51zVF)  and FastGen (https://openreview.net/forum?id=uNrFpDPMyo) both observe the attention patterns and use it to compress KV cache.

- Missing important work in both related work and baseline comparison. The paper did compare with StreamingLLM, but does not discuss it in related work. In fact, the paper misses many important KV cache prior works:
(1) Model Tells You What to Discard: Adaptive KV Cache Compression for LLMs, ICLR 2024, https://openreview.net/forum?id=uNrFpDPMyo
(2) SnapKV: LLM Knows What You are Looking for Before Generation, https://arxiv.org/abs/2404.14469
(3) XC-Cache: Cross-Attending to Cached Context for Efficient LLM Inference, https://arxiv.org/abs/2404.15420
(4) Layer-Condensed KV Cache for Efficient Inference of Large Language Models, https://arxiv.org/abs/2405.10637
(5) PyramidInfer: Pyramid KV Cache Compression for High-throughput LLM Inference, https://arxiv.org/abs/2405.12532
(6) PyramidKV: Dynamic KV Cache Compression based on Pyramidal Information Funneling, https://arxiv.org/abs/2406.02069


- Many writing sections are unclear.
(1) The introduction (line 55) says prior work has two limitations, does IntelLLM have these two limitations too? What is the conflict between increased computational overhead and memory optimization? Line 83 says significant, how much memory is saved? Any speed gains?
(2) The analysis in Section 3.1 is weak without any evidence or citations, the two conclusions are not convincing either.  For example, line 172 says they compromise the robustness of the attention distribution. What is the robustness of attention in the first place? And why would the covariates compromise this?  Line 180 says sliding window fails to reason effectively about long texts, any evidence or citation? Line 183 says they contribute to the collapse of the LLM, again, no evidence or justification.
(3) The two theorems in sections 3.2 and 3.3 are not theorems, and the paper provides no proof.

- Evaluation is weak and flawed. Table 1 (line 399) presents the results of IntelLLM on longbench. But unclear why the window size is 4K for IntelLLM and 8K for others. There is no side-by-side efficiency comparison either. Line 443 says the latency increased by 2.63% but the memory saves 50%, is it the case that IntelLLM always saves 50% memory? There is no ablation study on that.

**Questions:**

See weaknesses.

---

### Note · Authors · 2024-11-26

I have read and agree with the venue's withdrawal policy on behalf of myself and my co-authors.